# Genomic diversity of *Neisseria gonorrhoeae* Isolates in Kenya revealed by MLST, NG-MAST, and NG-STAR typing

Mary Wandia Kivata[1]*, Fredrick Lunyagi Eyase[2], Wallace Dimbuson Bulimo[2], Valerie Oundo[2], Esther Waguche[2], Wilton Mwema Mbinda[3], Margaret Wanjiku Mbuchi[2]*

**1** Department of Biological and Physical Sciences, Karatina University (KarU), Karatina, Kenya, **2** Kenya Medical Research Institute (KEMRI), Nairobi, Kenya, **3** Department of Chemistry and Biochemistry, Pwani University (PU), Mombasa, Kenya

* marywandia086@gmail.com, mkivata@karu.ac.ke (MWK); MMbuchi@kemri.go.ke (MWM)

## Abstract

### Background

Surveillance of *Neisseria gonorrhoeae* strains, their antimicrobial resistance (AMR) profiles, and transmission dynamics is essential in the prevention and control of gonococcal infections. In Kenya, gonococcal molecular surveillance remains limited, leaving gaps in understanding circulating sequence types (STs). This study characterized Kenyan *N. gonorrhoeae* isolates using multiple molecular typing schemes.

### Methods

Illumina MiSeq generated paired-end sequence reads prepared from 35 *N. gonorrhoeae* isolates recovered from males and females from four different regions in Kenya were analyzed. Assemblies were analyzed using PubMLST tools for multilocus sequence typing (MLST) and the *N. gonorrhoeae* sequence typing for antimicrobial resistance (NG-STAR) scheme. Multi-antigen sequence typing (NG-MAST) was carried out using the NG-MAST database. Phylogenetic relationships were assessed using concatenated NG-STAR loci and core genome-based analyses.

### Results

Twenty-two MLST STs were identified, including eight novel STs; ST-1932 was most frequent. NG-MAST revealed 29 STs, of which 26 were novel, with a newly described ST-19168 predominating in Nyanza region. NG-STAR identified 23 STs with variation across *mtrR*, *penA*, 23S *rRNA*, *gyrA*, *parC*, *ponA*, and *porB*. Phylogenetic analyses showed clustering of isolates into distinct groups with diverse AMR profiles. One cluster comprised isolates resistant to tetracycline and ciprofloxacin. No clear association was observed between MLST, NG-MAST, or NG-STAR types and specific AMR patterns.

**Data availability statement:** The sequence data generated in this study are publicly available. Raw sequence reads have been deposited in the NCBI BioProjects PRJNA481622 and PRJNA590515, accessible at https://www.ncbi.nlm.nih.gov/bioproject/PRJNA481622 and https://www.ncbi.nlm.nih.gov/bioproject/PRJNA590515 respectively. Genome assemblies are accessible via PubMLST at https://pubmlst.org/. Specific accession numbers for genome assemblies within PubMLST are provided in the S1 Table. All other relevant data supporting the findings of this study are contained within the manuscript and its supplementary files. For any inquiries regarding access to additional supporting data, please contact: Kenya Medical Research Institute (KEMRI) Scientific and Ethics Review Unit (SERU) P.O. Box 54840–00200, Off Mbagathi Road, Nairobi, Kenya House Number 8, KEMRI Headquarters Office Mobile: +254 717 719 477 Extension: 3333 or 3332 Email: seru@kemri.go.ke; kemriseru18@gmail.com.

**Funding:** The author(s) received no specific funding for this work.

**Competing interests:** The authors have declared that no competing interests exist.

## Conclusion

Kenyan *N. gonorrhoeae* strains are genetically diverse, with high numbers of novel NG-MAST and MLST STs. The lack of regional clustering and varied AMR profiles suggests widespread transmission of heterogeneous gonococcal populations. These findings underscore the importance of strengthened genomic surveillance to inform gonorrhea control strategies in Kenya.

## Introduction

Management of gonococcal infections is complicated by the emergence and spread of *N. gonorrhoeae* strains which are resistant to the current antibiotics recommended for treatment [1]. In addition, prevention and control of infections is hindered by the lack of a gonococcal vaccine [2]. Identification and genotypic characterization of these antibiotic resistant strains through molecular typing is a key factor in monitoring the emergence, transmission and spread of gonococcal drug resistance [3]. Characterization of sequence types and the associated antibiotic susceptibility profiles of circulating *N. gonorrhoeae* isolates provide epidemiological and drug resistance data useful for management and prevention of gonococcal infections including updating of treatment guidelines and other preventive interventions.

Both DNA-based (genotypic) and non-DNA-based (phenotypic) typing methods have been developed and used to characterize gonococcal isolates. Phenotypic typing methods include: antimicrobial susceptibility testing (AST); auxotyping; and serotyping [3]. They are limited in terms of reproducibility and their ability to discriminate between strains. DNA-based typing methods are categorized into gel-based and sequence-based typing methods. Gel based typing methods employ gel electrophoresis to analyze DNA bands and include: ribotyping; restriction fragment length polymorphism (RFLP) and pulsed-field gel electrophoresis (PFGE); opa-typing; and PCR based typing [3]. DNA sequence-based typing methods entail the analysis of one or more genes and include; *Neisseria gonorrhoeae* Multi-antigen Sequence Typing (NG-MAST); Multi-locus Sequence Typing (MLST); and *N. gonorrhoeae* Sequence Typing for Antimicrobial Resistance (NG-STAR) [4,5]. DNA sequence-based typing methods are preferred because: they have high discrimination power; are rapid and reproducible; allow identification of novel polymorphisms and also transfer and comparison of results [3].

MLST uses partial sequence information of seven relatively conserved, slow evolving housekeeping genes which are distributed throughout the genome (*abcZ*, *adk*, *aroE*, *fumC*, *gdh*, *pdhC*, and *pgm*) to assign an isolate to a sequence type [5,6]. It is recommended for typing gonococcal infections in long-term and global epidemiology studies [3]. NG-MAST analyses partial DNA sequences from hypervariable regions of two outer membrane proteins; porin B encoded by *porB* (490 pb) and, transferrin binding protein encoded by *tbpB* (390 bp). It uses an open access database (http://www.ng-mast.net/) for analysis and is recommended for typing gonococcal infections in micro-epidemiological studies [3, 4]. NG STAR is a newly described antimicrobial

resistance multi-locus typing scheme which is based on gonococcal antimicrobial resistance determinants. It uses alleles from seven genes associated with resistance to β-lactams, macrolides, and fluoroquinolones antibiotics. The genes include: *penA*; *mtrR*; *porB*; *ponA*; *gyrA*; *parC*; and 23S rRNA. This typing scheme provides insights on chromosomal antimicrobial determinants and is useful in monitoring global dissemination of antimicrobial resistant gonococcal strains [7].

Gonococcal antimicrobial susceptibility testing combined with DNA sequence-based typing data provide information useful in: understanding strain evolution and genetic relatedness; monitoring changes in existing strains and emergence of new drug resistant strains; and monitoring the spread of antibiotic resistant strains in specific groups or populations [3,8]. In Kenya few studies have characterized *N. gonorrhoeae* whole genome sequence data [9,10]. Consequently, data on the current circulating gonococcal sequence types is limited. This study used the commonly used and recommended DNA sequence-based typing methods to characterize genetic heterogeneity of *N. gonorrhoeae* isolates obtained from different regions in Kenya.

## Materials and methods

### Study isolates

Study isolates consisted of 35 *N. gonorrhoeae* isolates obtained from both male and female patients seeking treatment in selected clinics from four different regions in Kenya: Nairobi; Coastal Kenya; Nyanza (including the Kisumu and Kombewa study sites); and Rift Valley between 2013 and 2018. The isolates were collected and antimicrobial susceptibility testing done as part of an ongoing STI surveillance study titled "A surveillance study of antimicrobial susceptibility profiles of *N. gonorrhoeae* isolates from patients seeking treatment in selected clinics in Kenya" (WRAIR#1743, KEMRI#1908). Minimum inhibitory concentrations for: ceftriaxone; cefixime; penicillin; tetracycline; azithromycin; ciprofloxacin and specinomycin; (S1 Table) [11] were determined using E-test® (Biomerieux) method according to manufacturer's instructions [12,13] and the breakpoints interpreted with reference to European Committee on Antimicrobial Susceptibility Testing (EUCAST) version 8.0, 2018 standards (S2 Table). DNA isolation and whole genome sequencing was done as part of a retrospective laboratory based molecular sub-study titled "Molecular characterization of antimicrobial resistance genes in *Neisseria gonorrhoeae* isolates from Kenya through whole genome sequencing" which was nested in the STI surveillance programme. Genomic DNA was extracted using QIAamp DNA Mini Kit (QIAGEN, Hilden, Germany) and the quality and quantity of DNA determined by Qubit dsDNA HS Assay using Qubit 3.0 fluorometer, (Thermo Fisher Scientific Inc. Wilmington, Delaware USA) according to the manufacturer's instructions.

### Whole genome sequence data, assembly and annotation

Illumina Nextera XT kit (Illumina Inc. San Diego, CA, USA) was used to prepare libraries as per manufacturer's instructions. Sequence reads were generated on Illumina MiSeq platform (Illumina, San Diego, CA, USA) using a paired-end 2 × 300 bp protocol [14]. Sequence reads were assembled with CLC Genomics Workbench (v12.0, Qiagen). Genome annotation and analysis were conducted with BIGSdb tools accessed through the PubMLST platform (www.pubmlst.org/neisseria). The sequence data generated in this study are publicly available. Raw sequence reads have been deposited in the NCBI BioProjects PRJNA481622 and PRJNA590515, accessible at https://www.ncbi.nlm.nih.gov/bioproject/PRJNA481622 and https://www.ncbi.nlm.nih.gov/bioproject/PRJNA590515 respectively. Genome assemblies are accessible via PubMLST at https://pubmlst.org/. Specific accession numbers for genome assemblies within PubMLST are provided in the S3 Table.

### Multi-locus Sequence Typing, NG-MAST and NG STAR

Identification of multi-locus sequence types (MLST) was performed on assembled genome sequences using MLST version 1.8 (previously hosted at https://cge.cbs.dtu.dk/services/MLST/) available online at the Centre for Genome

Epidemiology (CGE) [15]. At the time of analysis, this platform provided a streamlined workflow for batch processing and generated allele definitions directly comparable to PubMLST, ensuring consistency and interoperability of sequence type assignments. MLST typing is now supported through the updated CGE platform (https://cge.food.dtu.dk/services/MLST/) or directly via PubMLST.

A local BLAST search against NG-MAST *porB* and *tbpB* databases was used to identify *porB* and *tbpB* genes from the assembled genomes using BioEdit sequence alignment editor version 7.0.5 [16]. Determination of *porB* and *tbpB* alleles and NG-MAST sequence types (STs) was originally performed using the NG-MAST website (www.ng-mast.net), which at the time was the standard resource for allele assignment. NG-MAST typing is now maintained within PubMLST, and future analyses should use the PubMLST-hosted NG-MAST scheme to ensure sustainability and consistency of results.

Determination of *N. gonorrhoeae* Sequence Typing for Antimicrobial Resistance (NG-STAR) STs was carried out using the NG-STAR scheme implemented in PubMLST [7]. While PubMLST hosts a version of NG-STAR, the primary host for the database is https://ngstar.canada.ca/, which should be used for future analyses to access the most up-to-date allele definitions and sequence type assignments.

## Phylogeny analysis

To identify phylogenetic relationships among the study sequences, genome comparator tool hosted on pubmlst.org/neisseria was used to compare whole genome sequence and core genome-based phylogeny inferred using cgMLST *N. gonorrhoeae* v.1.0 scheme. The generated data was visualized using Inkscape. ITOL tool hosted on pubmlst.org/neisseria was used to create a neighbor-joining tree from concatenated seven NG-STAR loci nucleotide sequences. The core genome comparison data was annotated by NG-STAR, NG-MAST and MLST schemes. To analyze clustering of sequence types in global context, ITOL was used to generate a neighbor-joining tree based on NG-STAR, NG-MAST and MLST schemes. To provide regional and global context, 55 additional *N. gonorrhoeae* genome sequences were retrieved from PubMLST. These included 35 isolates from other African countries (Uganda, Malawi, South Africa, Ethiopia, Eritrea, Morocco, Ghana, Zambia, Lesotho, Madagascar, Rwanda, Burkina Faso and Cameroon) to capture regional genetic diversity, and a randomized sample of 20 non-African isolates representing major global lineages (such as ST-1901 and ST-1407). Selection was prioritized for available isolates with collection dates between 2010 and 2020 to align with the study period.

## Statistical analysis

All proportions are reported with 95% confidence intervals (CIs) using the binomial exact method. Statistical comparisons were performed using Mann–Whitney U tests for continuous variables and Fisher's exact tests for categorical variables. Given the small sample size and exploratory nature of this study, p-values were not corrected for multiple testing and should be interpreted with caution. We abstracted sex, age, marital status, and self-reported partner count from clinic records. For partner count, categories were collapsed to >2 vs ≤2/none due to sparse upper cells. Genomic novelty was indicated by the presence of a novel sequence type in any scheme (MLST, NG-MAST, NG-STAR).

## Ethical consideration

Permission to carry out the study for both the STI surveillance study and the nested molecular sub-study were obtained from Kenya Medical Research Institute (KEMRI) Scientific and Ethics Review Unit) and Walter Reed Army Institute of Research Institutional Review Board as (WRAIR#1743, KEMRI#1908) and (WRAIR#1743A, KEMRI#3385) "respectively". Consent to participate was not applicable for this study because it was retrospective laboratory based, used archived samples, and there was no interaction with human subjects. All data were fully anonymized before we accessed them and the research was qualified as not involving human subjects. The isolates were acquired in two batches on 23/1/2017 and 26/2/2018.

## Results

All the thirty-five *Neisseria gonorrhoeae* isolates were successfully included across all three typing schemes, with sequential isolate IDs reflecting the availability of quality-assured WGS data; no isolates were excluded after typing. The majority of isolates were recovered from Nyanza (n = 23, 65.7%), with smaller numbers from Nairobi (n = 5, 14.3%), Rift Valley (n = 4, 11.4%), and the Coast region (n = 3, 8.6%). Collection spanned six years (2013–2018), with the largest numbers obtained in 2015 and 2016 (n = 9 each), followed by 2017 (n = 7), 2018 (n = 4), 2014 (n = 4), and 2013 (n = 2). The sample set was predominantly male (n = 29, 82.9%), with only six isolates recovered from female patients. Age distribution showed that most isolates were obtained from young adults, with the majority falling into the 20–29 and 30–39 age bands. Smaller proportions were observed among adolescents (18–19 age band) and older adults (40 + age band) (S1 Table).

## Sequence typing

### *N. gonorrhoeae* multi-locus sequence typing

A total of 22 MLST STs representing: 3 *abcZ*; 2 *adk*; 3 *aroE*; 6 *fumC*; 5 *gdh*; 3 *pdhC;* and 2 *pgm* different alleles were identified. Of the 35 sequences, 24 belonged to 14 known MLST STs; ST-1588; ST-1599; ST-1893; ST-1921; ST-1928; ST-1932; ST-8111; ST-8133; ST-11242; ST-11365; ST-11366; ST-11367; ST-11750; and ST-11976; while the remaining 11 sequences belonged to 8 new STs; ST-13613; ST-13614; ST-13763; ST-13764; ST-13766; ST-13779; ST-13780; and ST-13782 (S4 Table). The predominant ST was ST-1932 (n = 5, 14.3%) followed by ST-8133 (n = 4, 11.4%) and a novel ST-13782 (n = 3, 8.6%). In 2015, 44.4% (4/9) of the isolates displayed novel MLST types. Twenty isolates formed seven groups (two or more same ST) while 15 isolates formed singular MLST STs. Unlike ST-1932 and ST-13782, ST-8133 consisted of isolates recovered from one region; Nyanza. Region-based occurrence was also observed in ST-11365 and ST-13780 which consisted of two isolates each from Nyanza.

### *N. gonorrhoeae* multi-antigen sequence genotyping

In total, 29 NG-MAST sequence types were detected, encompassing 22 distinct *porB* alleles and 21 *tbpB* alleles. Nine of both the 22 *porB* and the 21 *tbpB* alleles were novel. Thirty-one isolates (88.6%) belonged to 26 novel NG-MAST STs, including ST-18599, ST-19087, ST-19166 through ST-19170, and ST-19254 through ST-19272. Four isolates (11.4%) belonged to three previously known STs: ST-355 (n = 1), ST-10134 (n = 2), and ST-11752 (n = 1). The most common ST overall was ST-19168, identified in four isolates (11.4%) from Nyanza. All four of these isolates also belonged to MLST ST-8133, which was the second most frequent MLST type observed. Sequence types ST-19255, ST-19262, and ST-10134 were each identified in two isolates, making them the second most common after ST-19168. In total, four ST groups (ST-19168, ST-19255, ST-19262, and ST-10134) accounted for 10 isolates, while the remaining isolates each represented unique NG-MAST STs (S5 Table). Regional clustering was observed only in two novel STs: ST-19168 and ST-19255, both of which comprised isolates recovered exclusively from Nyanza between 2014 and 2018.

### *N. gonorrhoeae* Sequence Typing for Antimicrobial Resistance (NG STAR)

Twenty-three NG-STAR STs representing 9 *mtrR*, 5 *penA*, 1 23S rRNA, 3 *gyrA*, 6 *parC*, 2 *ponA*, and 6 *porB* different alleles were identified. Seven (20%) isolates belonged to 6 novel NG-STAR STs: ST-3179; ST-3182; ST-3183; ST-3184; ST-3185 and ST-3186 while the rest of the isolates belonged to 17 known NG STAR STs. The most common STs were ST-1586 and ST-2668 which both comprised 4 (11.4%) isolates each. One of the two predominant NG-STAR ST, ST-2668 comprised isolates from Nyanza which also belonged to the most common NG-MAST, ST-19168 and to the second common MLST, ST-8133. Six isolates from Nyanza shared identical NG-STAR, NG-MAST, and MLST sequence types. Nineteen isolates formed 7 NG STAR ST groups while the rest formed singular NG-STAR STs (S6 Table)

## Temporal distribution of novel sequence types

Across 35 isolates analyzed between 2013 and 2018, the emergence of novel sequence types was observed across all three typing schemes. The yearly distribution of isolates was: 2013 (n = 2), 2014 (n = 4), 2015 (n = 9), 2016 (n = 9), 2017 (n = 7), and 2018 (n = 4). Due to the small sample sizes in 2013, 2014, and 2018, temporal trends in the proportion of novel sequence types should be interpreted with caution. Novel MLST STs were detected between 2014 and 2017, with the highest proportions observed in 2015–2016 (44.4% each). NG-MAST showed high rates of novelty throughout the study period, with novel STs identified in every year. While the proportion of novel NG-STAR STs appeared higher in 2017 (57.1%) and 2018 (75.0%), these figures represent a small number of isolates (4/7 and 3/4, respectively) and may not reflect broader population shifts (Fig 1). Collectively, these findings underscore the contrasting capacities of the three typing schemes: MLST captured moderate levels of novelty, NG-MAST identified extensive novel diversity, and NG-STAR revealed fewer but temporally clustered novel STs.

## Associations between epidemiological covariates and genomic novelty

Of the 35 cases (male 85.7%; median 20–29 years, IQR 22–31), "any novel sequence type" was common across demographic strata. No significant associations were detected between novelty and age (median 20–29 years vs. 30–39 years; U = 18.0, p = 0.322) or sex (male vs female: OR=7.25, p = 0.269). The high prevalence of novelty across strata suggests broad dissemination of novel lineages in the catchment population rather than concentration in specific demographic groups

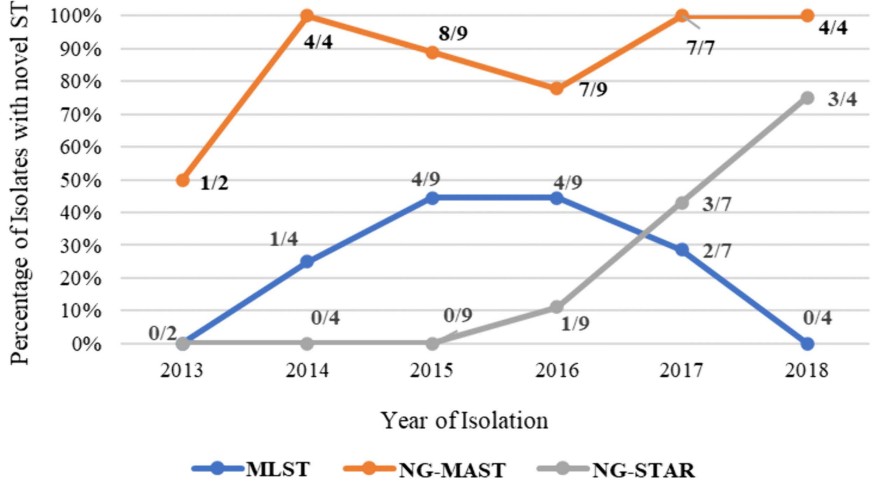

**Fig 1. Temporal trends for novel MLST, NG-MAST, and NG-STAR sequence types (2013–2018).** Percentage of isolates assigned to novel sequence types across MLST, NG-MAST, and NG-STAR typing schemes between 2013 and 2018. Bars represent the proportion of isolates with novel STs per year for each scheme (MLST in blue, NG-MAST in orange, NG-STAR in gray). MLST demonstrated moderate novelty, peaking in 2015–2016 (44.4%) but absent in 2018. NG-MAST revealed extensive novelty, with several years (2014, 2017, 2018) reaching 100%. NG-STAR exhibited the lowest overall novelty (20%), with novel STs concentrated in later years, particularly 2017 (42.9%) and 2018 (75%).

## Phylogenetic analyses

### Core genome-based phylogeny (cgMLST N. gonorrhoeae v.1.0 scheme)

Genome comparator tool identified 1485 loci as core genomes to *N. gonorrhoeae*. cgMLST *N. gonorrhoeae* v.1.0 scheme identified four clusters (group of more than 3 isolates) among the study isolates: Cluster 1 (n = 15); Cluster 2 (n = 6); Cluster 3 (n = 5) and Cluster 4 (n = 4). Cluster 1 was further sub-divided into two smaller but distinct sub-clusters; sub-cluster 1a (n = 9) and sub-cluster 1b (n = 6). Isolates in sub cluster 1b were from 3 regions and belonged to 3 different MLST, NG-MAST, and NG-STAR STs. Isolates in sub-cluster 1a belonged to: 5 MLST ST (3 novel and 2 existing STs), 8 NG_MAST STs (6 novel and 2 existing STs) and 4 NG-STAR STs. Eight of the nine isolates in this sub-cluster were all from Nyanza. Cluster 2 was formed by isolates from Nairobi and Nyanza belonging to: 2 existing MLSTs STs; 5 NG-MAST STs and 6 NG-STAR STs. Cluster 3 was formed by four isolates from Nyanza and one from Rift Valley. They belonged to; four different MLST and NG- MAST STs (all novel) and 2 NG-STAR STs. Isolates in cluster 4 do not form a true cluster but rather represent a set of genetically diverse strains, belonging to three existing MLST STs, four novel NG-MAST STs, and four NG-STAR STs, and were not as closely clustered as in the other three groups (Fig 2).

### NG-STAR phylogeny

Neighbor-joining tree inferred by ITOL identified 3 distinct clusters: Cluster A (n = 12); Cluster B (n = 9); and Cluster C (n = 6) (Fig 3). Cluster A was formed by 12 isolates from all the four sampled regions. These isolates had varied

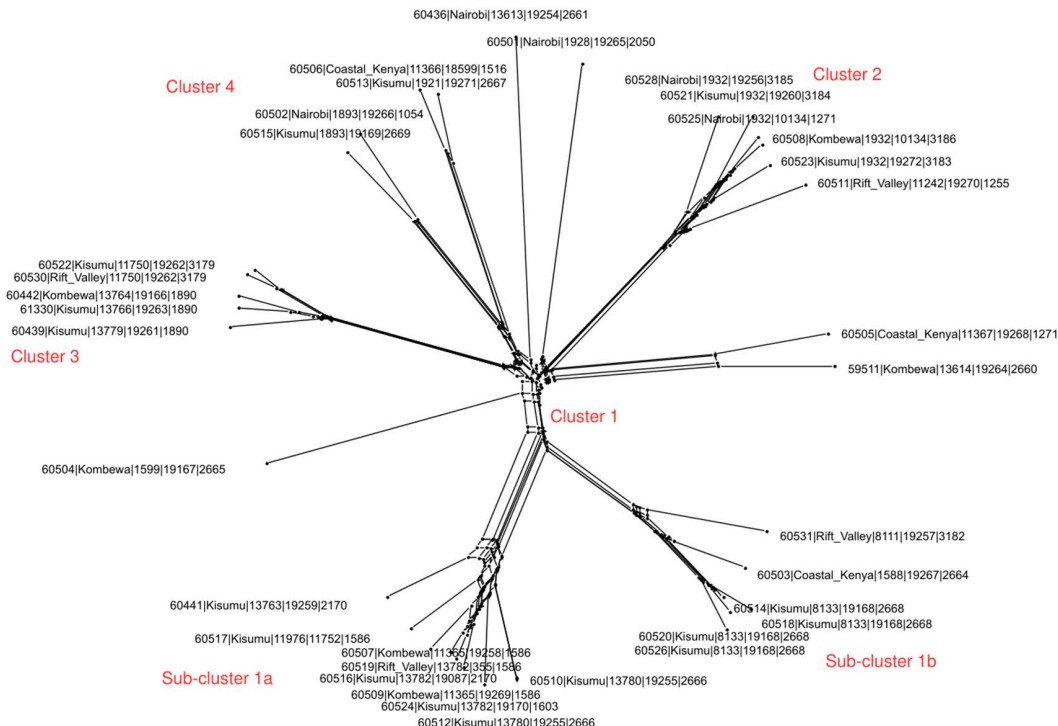

**Fig 2. Core genome-based phylogeny inferred from 1,485 loci identified as core in 35 Kenyan *Neisseria gonorrhoeae* isolates using the cgMLST *N. gonorrhoeae* v.** 1.0 scheme. Three distinct clusters (1-3) were formed; isolates designated as Cluster 4 do not represent a true cluster but instead comprise genetically diverse strains belonging to multiple sequence types. Shown in the leaf terminals are PubMLST IDs for the isolates, along with their MLST, NG-MAST, and NG-STAR sequence type assignments. For clarity, all isolate identifiers and sequence types are summarized in S3 Table.

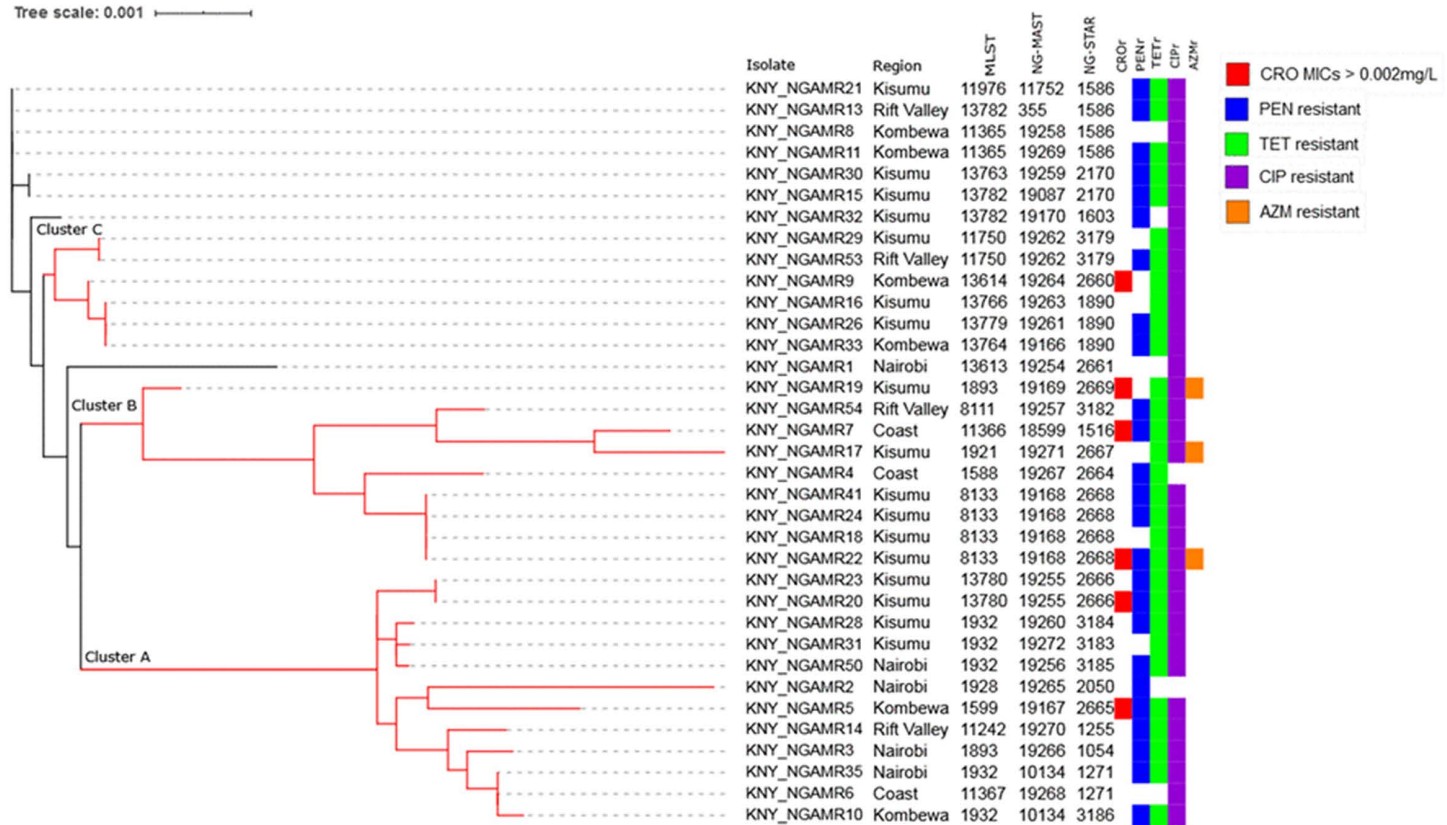

**Fig 3. Neighbor-joining tree generated by ITOL from concatenated sequences of seven NG-STAR loci (*mtrR, penA*, 23S rRNA, *gyrA, parC, ponA*, and *porB*) for 35 Kenyan *Neisseria gonorrhoeae* isolates.** Antimicrobial susceptibility profiles for isolates with reduced susceptibility to ceftri-axone (CRO; MICs > 0.002 mg/L) and those resistant to penicillin (PEN), tetracycline (TET), ciprofloxacin (CIP), and azithromycin (AZM) are indicated. NG-STAR sequence type labels are included to illustrate how specific STs cluster in relation to antimicrobial resistance profiles, complementing the phy-logeny inferred from concatenated loci. Three distinct clusters (A-C) comprised isolates with varied MLST, NG-MAST and NG-STAR STs. Branch lengths represent genetic distances between sequences.

antibiotic susceptibility profiles. Ten of these 12 isolates belonged to 6 already known MLST STs while the remain-ing 2 isolates belonged to a novel MLST ST. On the contrary more diversity was observed in this cluster with regard to NG-MAST and NG STAR STs as these isolates belonged to ten NG-MAST and ten NG-STAR STs. Nine of these ten NG-MAST STs formed by 10 isolates were novel while one ST formed by two isolates was an existing one (ST-10134).

The nine isolates forming cluster B were also from all the sampled regions and belonged to six existing MLST STs. The isolates belonged to six novel NG-MAST STs and six NG-STAR STs. They were all tetracycline resistant, while 8 of the 9 were ciprofloxacin resistant. Additionally, three isolates with a low-level resistance to azithromycin all belonged to this cluster. Cluster C was formed by six isolates; five from Nyanza and one from Rift Valley. Unlike cluster A and B which mostly consisted of existing MLST STs, cluster C isolates belonged to four novel MLST STs and one existing MLST ST. The isolates were all tetracycline and ciprofloxacin resistant. NG-STAR, NG-MAST and MLST region-based clustering was not observed in either of the two phylogenies as the identified clusters comprised isolates with varied STs. Resis-tance to ciprofloxacin, penicillin, and tetracycline was widespread across the three typing schemes (MLST, NG-MAST, NG-STAR).

## Clustering of Kenyan isolates in a global and regional context

To assess the genetic relationship of Kenyan *N. gonorrhoeae* isolates within a broader context, a neighbor-joining tree was constructed using MLST, NG-MAST, and NG-STAR schemes, incorporating 50 additional sequences from PubMLST (Fig 4). The updated analysis reveals that Kenyan isolates are distributed across multiple distinct clusters, showing both regional affinity and evidence of localized evolution. A significant proportion of Kenyan isolates clustered closely with sequences from other African countries, including Uganda, Malawi, South Africa, Cameroon, and Burkina Faso. Notably, Kenyan isolates with novel NG-MAST and NG-STAR types often branched within these African-centric clades, indicating that while these specific sequence types are newly described, they have emerged from established regional genetic backgrounds.

In contrast, the Kenyan isolates remained phylogenetically distinct from the major global multidrug-resistant (MDR) lineages. A randomized sample of non-African isolates, including those from Europe (UK, Spain, France), the Americas (USA, Brazil, Argentina), and Asia (Japan, China, Vietnam), predominantly clustered within well-known high-risk lineages such as MLST ST-1901 and NG-MAST ST-1407 [17,18]. None of the Kenyan isolates from this study period (2013–2018) clustered within these pandemic MDR clades, reinforcing the observation that the Kenyan gonococcal population during this time was characterized by locally and regionally evolving lineages rather than the introduction of global MDR clones.

## Discussion

The present study demonstrates substantial genetic diversity among Kenyan *Neisseria gonorrhoeae* isolates, with a high proportion of novel sequence types identified across MLST, NG-MAST, and NG-STAR schemes. Most NG-MAST and NG-STAR STs were newly described, underscoring independent evolution and ongoing diversification within the Kenyan gonococcal population. NG-MAST showed the greatest discriminatory power, with the majority of isolates forming unique STs, consistent with findings from other regions such as Western Australia and Japan [19,20].

Temporal analysis revealed dynamic population shifts: 2015–2017 marked peaks in genomic novelty across all schemes, while 2018 isolates showed fewer novel MLST types but continued emergence of novel NG-STAR STs. This

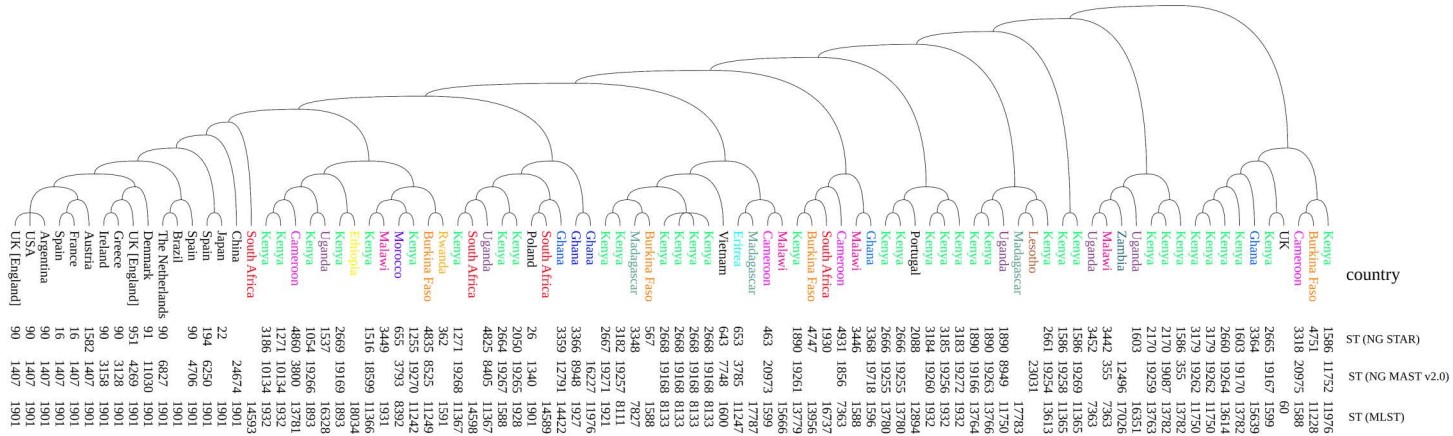

**Fig 4. Global and Regional Phylogenetic Context of Kenyan *N. gonorrhoeae* Isolates.** Neighbor-joining tree generated by ITOL using MLST, NG-MAST, and NG-STAR schemes (n = 90). The analysis includes 35 Kenyan isolates from the present study (2013−2018) and 55 additional genome sequences from PubMLST, comprising 35 regional African and 20 randomized global reference strains representing major multidrug-resistant (MDR) lineages. Kenyan isolates are distributed across multiple distinct clusters, showing strong regional affinity with other African sequences, particularly those from neighboring East African countries. In contrast, the Kenyan isolates remained phylogenetically distinct from the major global MDR lineages (ST-1901 and ST-1407), which predominantly clustered within non-African clades. PubMLST IDs, MLST, NG-MAST, and NG-STAR sequence types are shown in the leaf terminals.

suggests that even during periods of apparent lineage stabilization, resistance determinants continue to evolve. However, the apparent stabilization of lineages in 2018 must be interpreted cautiously, as the small sample size and the under-representation of Kenyan and broader African isolates in international MLST, NG-MAST, and NG-STAR databases likely contribute to the high proportion of novel STs observed. The presence of novel NG-STAR STs in 2017–2018, for example, is more plausibly explained by limited African representation in these databases rather than true lineage diversification alone. Consequently, robust inference of lineage stabilization or genomic diversity trends requires larger sample sizes and more comprehensive databases with sufficient regional representation. Expanding the inclusion of African gonococcal isolates in international repositories is therefore critical to accurately capture the true diversity of *N. gonorrhoeae*, contextualize local findings within global population structures, and enable reliable evaluation of temporal changes in genomic diversity worldwide.

Antimicrobial resistance was widespread, particularly to ciprofloxacin, penicillin, and tetracycline, which were detected across multiple genetic backgrounds. This diffuse distribution reflects long-standing circulation of resistance determinants and the role of horizontal gene transfer in shaping gonococcal populations. No statistically significant associations were observed between specific sequence types and resistance profiles. Nevertheless, the detection of low-level azithromycin resistance in isolates belonging to distinct sequence types emphasizes the potential for resistance to emerge independently across diverse lineages. These findings argue for integrated genomic-AMR monitoring that incorporates metadata such as sexual networks, travel history, and partner concurrency, which may provide finer resolution of transmission dynamics than routine demographic variables.

Our findings are further contextualized by a recent large-scale genomic study by Mehta et al. (2025), which analyzed 218 *N. gonorrhoeae* isolates from Kisumu, Kenya, spanning two decades (2002–2009 and 2020–2022) [21]. Similar to our observations, Mehta et al. reported high levels of genetic diversity and the widespread prevalence of resistance to ciprofloxacin, penicillin, and tetracycline across both time periods. Notably, several sequence types identified in our study, such as MLST ST-8133 and ST-1932, were also prominent in their collection, suggesting the persistence and broad dissemination of these lineages within Kenya. However, some key differences in antimicrobial resistance trends were noted. While our 2013–2018 isolates showed only low-level azithromycin resistance and remained susceptible to extended-spectrum cephalosporins, Mehta et al. documented the emergence of azithromycin resistance (n = 2) and cephalosporin alert values (n = 5) in their 2020–2022 cohort. This suggests a recent shift in the AMR landscape in Kenya, potentially occurring after our study period. Furthermore, Mehta et al. utilized a novel LIN code nomenclature based on cgMLST, which provided high resolution for lineage tracking, complementing our use of MLST, NG-MAST, and NG-STAR. The congruence between these different typing schemes across both studies underscores the robustness of genomic surveillance in capturing the dynamic evolution of *N. gonorrhoeae* in the region.

Phylogenetic analyses revealed incongruences between core genome-based and NG-STAR phylogenies. While cgMLST captured long-term evolutionary relationships, NG-STAR reflected recent adaptive events driven by antibiotic pressure. These differences highlight the complementary value of both approaches: cgMLST for broader lineage tracking and NG-STAR for monitoring resistance gene evolution.

The inclusion of a more diverse set of African isolates in our phylogenetic analysis provides a clearer understanding of the evolutionary trajectory of *N. gonorrhoeae* in Kenya. The close clustering of Kenyan isolates with those from neighboring African countries highlights the importance of regional transmission networks in the dissemination of gonococcal lineages across Africa. The persistence of certain sequence types, such as MLST ST-8133 and ST-1932, across multiple countries in the region suggests that these lineages are well-adapted to the local environment and may be contributing to the stable prevalence of legacy resistance (ciprofloxacin, penicillin, and tetracycline) observed in these populations.

Our findings also underscore the value of regional genomic surveillance. By comparing our data with a broader African dataset, we were able to demonstrate that the high number of novel sequence types identified in our study does not necessarily represent a radical departure from known strains, but rather a continuous diversification within established African

lineages. This is consistent with the findings of Mehta et al. (2025), who also noted the presence of both globally distributed and Africa-specific clades in their longitudinal study in Western Kenya [21].

The distinct separation between Kenyan isolates and global MDR lineages is particularly noteworthy. While these pandemic clones have driven treatment failures in many parts of the world, our data suggests they had not yet established a significant presence in the Kenyan regions sampled between 2013 and 2018. This may explain the continued effectiveness of extended-spectrum cephalosporins in Kenya during this period. However, the more recent emergence of azithromycin resistance and cephalosporin alert values reported by Mehta et al. (2020–2022) serves as a critical warning that the situation is dynamic and that the introduction or local evolution of more resistant lineages is an ongoing threat.

Although the global MDR lineages have not been detected in Kenya, regional and international travel could facilitate rapid importation. Targeted sentinel surveillance among border towns, mobile populations, and key populations (including MSM and sex workers), integrated into WHO GASP and aligned with the Kenya AMR Action Plan, is recommended to strengthen preparedness and response.

Generally, region-based clustering was not observed in either of the phylogenies. Although the study sample size is small, these observations suggest that the Kenyan gonococci sampled belong to a heterogeneous population. Nevertheless, sequence type-based clustering was observed in three schemes: MLST (ST-11750, ST-13780, and ST-8133), NG-MAST (ST-19262, ST-19255, and ST-19168), and NG-STAR (ST-3179, ST-2666, and ST-2668), predominantly among isolates from Nyanza. The apparent regional restriction of these sequence types should be interpreted with caution, as the larger number of isolates obtained from Nyanza compared to other regions likely contributed to their detection in that region alone. This uneven distribution of isolates highlights the need for more balanced sampling across regions to accurately assess lineage distribution and transmission dynamics.

Although this study provides valuable insights into the molecular epidemiology of *N. gonorrhoeae* in Kenya, the relatively small sample size (n = 35) collected over six years limits statistical power and generalizability. Rare but epidemiologically important sequence types or resistance profiles may not have been captured, and observed proportions of novel STs may be affected by sampling variability. Metadata were restricted to basic demographics, with under-representation of females and sparse reporting of high-partner categories, reducing the ability to detect epidemiological correlates of genomic novelty. Future surveillance should incorporate enhanced metadata, including sexual network information and travel history, to better capture transmission dynamics. The temporal and geographic representativeness of the dataset was also constrained. Although isolates were obtained from four regions, sampling was uneven across areas and years. The absence of certain high-risk clones such as ST-1901 and ST-1407 cannot be interpreted as definitive evidence of their absence in Kenya. Robust detection of low-prevalence but clinically significant genotypes require larger, systematic sampling across sentinel sites, with annual isolate counts sufficient to detect sequence types at low prevalence with adequate statistical power. Such efforts would enable reliable temporal trend analyses, phylogeographic inferences, and real-time detection of emerging AMR-associated clones.

## Conclusions and recommendations

Kenyan *N. gonorrhoeae* isolates are genetically diverse, with many novel sequence types identified across MLST, NG-MAST, and NG-STAR schemes. Resistance to ciprofloxacin, penicillin, and tetracycline is widespread and diffuse across multiple genetic backgrounds, reflecting long-standing circulation of resistance determinants. The emergence of novel NG-STAR STs underscores the ongoing evolution of resistance genes. Although limited by sample size, the absence of globally dominant high-risk clones such as MLST ST-1901 and NG-MAST ST-1407 provides an opportunity to strengthen local containment before their introduction.

We recommend strengthening genomic surveillance, integrating resistance data into Kenya's AMR action plan, establishing sentinel sites with systematic sampling, and enhancing collaboration between laboratories, public health institutions, and clinicians to ensure timely feedback into patient management. These measures are critical to prevent the spread of highly resistant strains and safeguard current treatment regimens.

## Supporting information

**S1 Table. Isolate metadata and antimicrobial susceptibility data.**
(DOCX)

**S2 Table. EUCAST v8.0 (2018) Breakpoints for *N. gonorrhoeae*.**
(DOCX)

**S3 Table. Sequential numbering (1–35) of the study isolates with their corresponding PubMLST identifiers and assigned sequence types across the three typing schemes (MLST, NG-MAST, and NG-STAR).** This table provides a cross-reference for isolates shown in Figure 2, allowing readers to easily match strain numbers to the PubMLST IDs and sequence type designations. Novel sequence types (STs) identified in this study are indicated in bold italics.
(DOCX)

**S4 Table. Identified MLST alleles and sequence types.** *Novel sequence types (STs) identified in this study are indicated in bold italics.*
(DOCX)

**S5 Table. Identified NG-MAST alleles and sequence types.** *Novel sequence types (STs) identified in this study are indicated in bold italics.*
(DOCX)

**S6 Table. Identified NG-STAR alleles and sequence types.** *Novel sequence types (STs) identified in this study are indicated in bold italics.*
(DOCX)

## Author contributions

**Conceptualization:** Mary Wandia Kivata, Fredrick Lunyagi Eyase, Wallace Dimbuson Bulimo, Valerie Oundo, Esther Waguche, Wilton Mwema Mbinda, Margaret Wanjiku Mbuchi.

**Data curation:** Mary Wandia Kivata.

**Formal analysis:** Mary Wandia Kivata, Wallace Dimbuson Bulimo, Margaret Wanjiku Mbuchi.

**Funding acquisition:** Margaret Wanjiku Mbuchi.

**Investigation:** Mary Wandia Kivata, Fredrick Lunyagi Eyase, Wallace Dimbuson Bulimo, Valerie Oundo, Esther Waguche.

**Methodology:** Mary Wandia Kivata, Fredrick Lunyagi Eyase, Wallace Dimbuson Bulimo, Valerie Oundo, Wilton Mwema Mbinda, Margaret Wanjiku Mbuchi.

**Project administration:** Mary Wandia Kivata, Fredrick Lunyagi Eyase, Margaret Wanjiku Mbuchi.

**Resources:** Fredrick Lunyagi Eyase, Wallace Dimbuson Bulimo, Margaret Wanjiku Mbuchi.

**Supervision:** Fredrick Lunyagi Eyase, Wallace Dimbuson Bulimo, Wilton Mwema Mbinda, Margaret Wanjiku Mbuchi.

**Visualization:** Mary Wandia Kivata.

**Writing – original draft:** Mary Wandia Kivata, Wallace Dimbuson Bulimo.

**Writing – review & editing:** Mary Wandia Kivata, Fredrick Lunyagi Eyase, Wallace Dimbuson Bulimo, Valerie Oundo, Esther Waguche, Wilton Mwema Mbinda, Margaret Wanjiku Mbuchi.

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
