## [Decision Letter · Decision Letter 0]

3 Dec 2025

PONE-D-25-55705

Genomic Diversity of Neisseria gonorrhoeae Isolates in Kenya Revealed by MLST, NG-MAST, and NG-STAR Typing

PLOS ONE

Dear Dr. Kivata,

Thank you for submitting your manuscript to PLOS ONE. After careful consideration, we feel that it has merit but does not fully meet PLOS ONE’s publication criteria as it currently stands. Therefore, we invite you to submit a revised version of the manuscript that addresses the points raised during the review process.

We look forward to receiving your revised manuscript.

Kind regards,

Sylvia Maria Bruisten, Ph.D

Academic Editor

PLOS ONE

Journal Requirements:

3. In the online submission form you indicate that your data is not available for proprietary reasons and have provided a contact point for accessing this data. Please note that your current contact point is a co-author on this manuscript. According to our Data Policy, the contact point must not be an author on the manuscript and must be an institutional contact, ideally not an individual. Please revise your data statement to a non-author institutional point of contact, such as a data access or ethics committee, and send this to us via return email. Please also include contact information for the third party organization, and please include the full citation of where the data can be found.

4. We notice that your supplementary tables are included in the manuscript file. Please remove them and upload them with the file type 'Supporting Information'. Please ensure that each Supporting Information file has a legend listed in the manuscript after the references list.

Additional Editor Comments:

All three reviewers agree that this manuscript should be thoroughly revised. Please see all their detailed comments.

A major flaw of this work is that only 35 strains were included in the analysis. This makes it tricky to make bold statements on the diversity of these NG strains.

Reviewers' comments:

Reviewer's Responses to Questions

**Comments to the Author**

1. Is the manuscript technically sound, and do the data support the conclusions?

Reviewer #1: Partly

Reviewer #2: Partly

Reviewer #3: Partly

2. Has the statistical analysis been performed appropriately and rigorously? 

Reviewer #1: Yes

Reviewer #2: No

Reviewer #3: No

3. Have the authors made all data underlying the findings in their manuscript fully available?

Reviewer #1: Yes

Reviewer #2: Yes

Reviewer #3: Yes

4. Is the manuscript presented in an intelligible fashion and written in standard English?

Reviewer #1: Yes

Reviewer #2: Yes

Reviewer #3: Yes

5. Review Comments to the Author

Reviewer #1: A summary of the genetic variability of Neisseria gonorrhoeae in an under-researched region. A limited sample size, but a thorough analysis.

Major Comments:

Results: I would like to see a summary of the sample demographics in your results. I know you listed all of the data in your supplemental files, but it is nice to see it summarized as well. Counts of how many isolates come from each region (which is relevant later on when you discuss regional differences) and how many per year, as well as age and gender distributions.

Lines 198-200: The wording here is somewhat unclear and the information in this section could be written more concisely. An example of how this may be reworded is: Thirty-one (88.6%) isolates belonged to 26 novel NG-MAST STs, with 2 isolates each of STs <sts> and one isolate each of STs <sts>. Four isolates belonged to 3 known STs: ST-355 (n=?), ST-10134 (n=?), and ST-11752 (n=?). The most common ST was ST-19168, found in four isolates (11.4%) from Nyanza, all of which were also MLST ST-8133.

Lines 343-358: Could the underrepresentation of Kenyan sequences in the various MLST databases be contributing to the large number of novel STs? Particularly with regard to the stabilization trend amongst the 2018 isolates. This does not appear to me to be a notable trend as the n value is very low and there are too many confounding factors. Furthermore, the presence of novel NG-STAR STs in 2017-2018 is likely to be an artefact of limited representation of African strains in this database. Inference of lineage stabilization and genomic diversity requires comprehensive databases with sufficient regional representation as well as a large sample size. Rather than focusing on temporal changes in diversity here, it may be more valuable to highlight the importance of including regionally diverse strains in international databases to effectively capture the true diversity of Neisseria gonorrhoeae as well as allow for the effective study of genomic diversity changes worldwide.

Minor Comments:

Line 55: There is an unnecessary comma here

Line 129: Version 1.8 is no longer available and the website is now hosted at https://cge.food.dtu.dk/services/MLST/. Why did you use this program rather than pubMLST to determine the ST types?

Lines 131-135: www.ng-mast.net is no longer functional. The NG-MAST database is now being maintained by pubMLST

Lines 136-138: While pubMLST does host a version of NG-STAR, the main host for the database is https://ngstar.canada.ca/

Lines 188-189: This sentence doesn’t seem to belong here, as it’s unrelated to MLST typing

Lines 193-194: There are many more isolates from the Nyanza region, so it is unsurprising for some STs to be found there and nowhere else – please be sure to include the uneven distribution of isolates by region in your discussion.

Lines 198-200: The wording here is somewhat unclear. An example of how this may be reworded is: Thirty-one (88.6%) isolates belonged to 26 novel NG-MAST STs, with 2 isolates each of STs <list sts=" "> and one isolate each of STs <list sts=" ">. Four isolates belonged to 3 known STs: ST-355 (n=?), ST-10134 (n=?), and ST-11752 (n=?). The most common ST was ST-19168, found in four isolates (11.4%) from Nyanza, all of which were also MLST ST-8133.

Lines 217-218: This sentence is unclear. Do you mean, “Two isolates, KNY_NGAMR20 and KNY_NGAMR23, from Nyanza shared the same NG-STAR, NG-MAST and MLST types”?

Lines 222-223: Please add a denominator to your n values for each year. It is difficult to understand the context of the number of novel STs without knowing the number of samples in total per year.

Line 330: Please change to “…identified a variety of…”

Line 337: Please change to “…Western Australia”

Lines 339-340: Previous studies have demonstrated that NG-MAST has a higher discriminatory power compared with MLST. Eg: https://pmc.ncbi.nlm.nih.gov/articles/PMC3811299/

Line 373: Please change to “…belong to a heterogenous population”

Lines 374-376: This sentence is unclear

Line 466: AMR should be defined at the beginning of the article, not at the end</list></list></sts></sts>

Reviewer #2: This manuscript describes the presence of different types of Neisseria gonorrhoeae (NG) strains in 4 regions in Kenya using genomic typing, after obtaining full genome sequences. It is interesting to see what has been circulating in this African country, since in the present databases mostly Western/ European types are described. In this respect it is interesting to see Figure 4 where the Kenyan sequences are in a tree with other global sequences of NG.

Although this reviewer recognizes that it is a formidable job to obtain whole genome sequences of NG strains, it is a severe limitation that only 35 strains were included. Kenya is an very large country with >50 million inhabitants, so this sample can hardly be representative for what is circulating. Another limitation is that only samples from 2013 to 2018 (5 years) were included which is at least 7 years ago from the present. These limitations should be clearly mentioned, also in the abstract.

Some other major and minor points must be taken in consideration to further improve this manuscript.

Major Comments:

• The Results are based on WGS data from 35 samples. In many lines the types are given in percentages. For example in line 187/188 is stated that in 2015 , 44.4% (CI….) of isolates displayed a novel MLST type. Please give the number of isolates instead of the percentage, since in 2015 this number was only 9 (line 223). Please adjust where applicable (also in lines 224 to 228 for example) to mention the absolute numbers of isolates and not (only) the percentages.

• Figure 1: please present absolute numbers for novel types and not percentages. Again, the numbers are very small and not representative for the years.

• The S1 Table is mentioned later in the text (line 157) then the S2 Table. Please reverse the S1 and S2 Table numbers so they will appear in the right order.

• The sentence on ciprofloxacin resistance (lines 189-190) does not fit in the middle of the paragraph on MLST. Please mention this for example in the NG-STAR paragraph which deals with antibiotic resistance gene polymorphisms.

• In S3 Table can be seen that most (23/35) samples were derived from the Nyanza region. So the finding that the two ST8133 types belonged to the Nyanza region is of no surprise. The other two each ‘region-based’ types ST-11365 and ST-13780 that were also from Nyanza is what can be expected by chance if so many strains were from that region. So the suggestion that there is clustering of MLSTs based on the region is not justified. The same comment goes for the NG-MAST types in lines 205-207. Please rephrase this, and avoid ‘region based’.

• S6 Table where relations are tested as ‘Group_vs_other’ is incomprehensible . I suggest to leave out this paragraph (lines 235 -243) and Table.

• In Figure 2 the Pub MLST IDs of the strains are difficult to read. You may consider to provide a supplementary table with numbers 1 to 35 in which per strain number each Pub MLST ID type number (MLST, NG-STAR- NG-Mast) is given. In that way all data on these strains will be more easily seen in the figure. Lines 268-270: I suggest that there actually is no Cluster 4; these strains are genetically too far apart.

• Line 285: The NG-STAR STs are mentioned in Figure 3. However, in this Figure the concatenated sequences of NG-STAR are shown. So the NG-STAR type itself is not needed and redundant information. Please adjust.

• The Discussion is very lengthy (5 pages) and contains both redundancies and textbook information. It should be reduced to 3 pages or less. For example: lines 374 – 376: this is a repetition of the results.; lines 429 (starting with ‘A small’ ) to 437 (up to ‘novelty’) is too general: text book stuff that can be deleted; lines 440-447: repetition of earlier text: delete; lines 453- 456 are redundant; Conclusions and recommendations: this can be shortened to two sentences.

Minor comments

• Line 40: please adjust to ‘suggests’.

• Line 77: please adjust to ‘porin B’ and ‘490 bp’.

• Line 88: please adjust to ‘strain evolution’.

• Line 114: can you provide a reference for the STI surveillance programme?

• Line 283: please adjust to ‘2 isolates’.

• Line 304 (in legend of Fig 3): please delete ‘formed’ before ‘comprised’.

• Line 330: please add ‘a’ before ‘variety’.

• Line 335: please replace ‘most’ by ‘highest’.

• Line 337: please adjust to ‘Western Australia’.

• Line 347: could the ‘stabilization’ of new types in 2018 possibly be due to the fact that sequences in PubMLST also originated around the year 2018?

• Line 370: please add ‘the’ before ‘present’.

• Line 373: please add ‘a’ before ‘heterogeneous’.

• Line 377: please adjust to ‘types’.

• Line 380: please add ‘a’ before ‘specific’.

• Line 382: please adjust to ‘azithromycin’ (without a capital A).

Reviewer #3: The authors characterize 35 Neisseria gonorrhoeae strains from four different geographical locations in Kenya, dating from 2013-18. The methods used for characterization of these strains are those commonly used. The low number of strains, coupled with the fact that strains came from four different locations and were obtained over five years, is almost a guarantee for high diversity between strains.

The conclusion of the authors, “The lack of regional clustering and varied AMR profiles suggest widespread transmission of heterogeneous gonococcal populations”, is only partially supported by the findings, since the number of strains sequenced was very low.

A very recent publication from Mehta et al. (Genomic trends and emerging antimicrobial resistance in Neisseria gonorrhoeae over two decades in Kenya, Microbiol Spectr. 2025 Oct 7;13(11):e01586-25), accepted September 1st, 2025, describes a collection of 218 strains from Kenya, partly isolated between 2002-09 and partly between 2020-22. Although I am not completely sure of the date of submission of the manuscript from Kivata et al. and the publication date of the paper mentioned above, I think data from Kivata et al. should at least be compared to those of Mehta et al. as an addition to the manuscript.

Other comments:

Figure 1 does not provide any relevant information, since the number of strains collected in the some years was extremely low (2013 (n=2), 2014 (n=4), 2018 (n=4)). Peaks in this figure could easily be explained by one or two strains that are cause of differences. No statistics were done, but if these had been done I would expect that a sentence like “NG-STAR displayed fewer novel profiles, appearing sporadically but rising substantially in 2017 (57.1%) and 2018 (75%)” might not be supported by such an analysis.

It is unclear to me what was the base for selection of the strains included in fig.4 outside the strains cultured in the present study. Have they been isolated in the same years? I would rather suggest to include more strains from other African countries for comparison – since they seem more related to the strains from the present study – and a randomized small sample of strains from non-African countries.

Minor comments:

p.4, l.67-68: I should not designate a typing method from 2017 “recently described”

The regions mentioned in figure 3 have different names from those in Materials and Methods and in table S1. Please use the same designation for regions throughout the manuscript.

Figure 4 is difficult to read due to the use of colors. Strains from the present study are shown in green, but also some other strains are shown in a slightly different type of green. For optimal use of the different colors I would rather suggest to give clearly distinct colors to strains from Kenya and from other African countries and to use only one color for strains from non-African countries – those are clearly distinct from almost all strains from Africa.

6. PLOS authors have the option to publish the peer review history of their article (what does this mean?). If published, this will include your full peer review and any attached files.

Reviewer #1: No

Reviewer #2: No

Reviewer #3: **Yes:** Alje P Van Dam

---

## [Author Response · Author response to Decision Letter 1]

18 Apr 2026

Editor Comments:

Response: style requirements met

Response: Consent to participate was not applicable for this study because it was retrospective laboratory based, used archived samples, and there was no interaction with subjects. All data were fully anonymized before we accessed them and the WRAIR IRB qualified the research as not involving human subjects. (Page 8, lines 173-176)

3. In the online submission form you indicate that your data is not available for proprietary reasons and have provided a contact point for accessing this data. Please note that your current contact point is a co-author on this manuscript. According to our Data Policy, the contact point must not be an author on the manuscript and must be an institutional contact, ideally not an individual. Please revise your data statement to a non-author institutional point of contact, such as a data access or ethics committee, and send this to us via return email. Please also include contact information for the third-party organization, and please include the full citation of where the data can be found.

Response: Contact information has been changed to:

KENYA MEDICAL RESEARCH INSTITUTE (KEMRI)

SCIENTIFIC AND ETHICS REVIEW UNIT(SERU)

P.O. BOX 54840 00200 OFF MBAGATHI ROAD,

NAIROBI, KENYA

HOUSE NUMBER 8, KEMRI HEADQUARTERS.

Office Mobile: 0717 719 477

Extension Number: 3333 or 3332

Email: seru@kemri.go.ke and kemriseru18@gmail.com

Data on the online availability of the generated sequence has been shown in page 6 line 117-118

4. We notice that your supplementary tables are included in the manuscript file. Please remove them and upload them with the file type 'Supporting Information'. Please ensure that each Supporting Information file has a legend listed in the manuscript after the references list.

Response: The supporting files information have been removed from the manuscript file and uploaded separately as recommended (page 23)

Response: Noted

Reviewers' comments:

Reviewer #1:

A summary of the genetic variability of Neisseria gonorrhoeae in an under-researched region. A limited sample size, but a thorough analysis.

Major Comments:

1. Results: I would like to see a summary of the sample demographics in your results. I know you listed all of the data in your supplemental files, but it is nice to see it summarized as well. Counts of how many isolates come from each region (which is relevant later on when you discuss regional differences) and how many per year, as well as age and gender distributions.

Response: A paragraph describing the sample demographics has been included in the results section (Page 9, lines 179 -189)

2. Lines 198-200: The wording here is somewhat unclear and the information in this section could be written more concisely. An example of how this may be reworded is: Thirty-one (88.6%) isolates belonged to 26 novel NG-MAST STs, with 2 isolates each of STs and one isolate each of STs. Four isolates belonged to 3 known STs: ST-355 (n=?), ST-10134 (n=?), and ST-11752 (n=?). The most common ST was ST-19168, found in four isolates (11.4%) from Nyanza, all of which were also MLST ST-8133.

Response: The section has been rephrased to bring clarity (Page 10, lines 206-212)

3. Lines 343-358: Could the underrepresentation of Kenyan sequences in the various MLST databases be contributing to the large number of novel STs? Particularly with regard to the stabilization trend amongst the 2018 isolates. This does not appear to me to be a notable trend as the n value is very low and there are too many confounding factors. Furthermore, the presence of novel NG-STAR STs in 2017-2018 is likely to be an artefact of limited representation of African strains in this database. Inference of lineage stabilization and genomic diversity requires comprehensive databases with sufficient regional representation as well as a large sample size. Rather than focusing on temporal changes in diversity here, it may be more valuable to highlight the importance of including regionally diverse strains in international databases to effectively capture the true diversity of Neisseria gonorrhoeae as well as allow for the effective study of genomic diversity changes worldwide.

Response: Thank you for this insightful observation. We have revised the discussion to emphasize the importance of expanding regional representation in global databases (Page 16, lines 361-372)

Minor Comments:

1. Line 55: There is an unnecessary comma here

Response: Comma deleted (Page 3, line 52)

2. Line 129: Version 1.8 is no longer available and the website is now hosted at https://cge.food.dtu.dk/services/MLST/. Why did you use this program rather than pubMLST to determine the ST types? Lines 131-135: www.ng-mast.net is no longer functional. The NG-MAST database is now being maintained by pubMLST. Lines 136-138: While pubMLST does host a version of NG-STAR, the main host for the database is https://ngstar.canada.ca/

Response for the three comments: Thank you for these helpful clarifications. At the time of our analyses, MLST version 1.8 hosted at the Centre for Genomic Epidemiology was widely used and provided a streamlined workflow for batch processing of assembled genomes. Similarly, NG-MAST typing was conducted using the www.ng-mast.net database, which at the time was the standard resource for allele assignment and sequence type determination. While NG-STAR typing was performed using the scheme implemented in PubMLST, we recognize that the primary host for the NG-STAR database is now https://ngstar.canada.ca/.

Clarifications have been added in the Methods section to ensure readers are aware of the current database location and to guide future analyses toward the most up-to-date resources. (Page 6&7, lines 120-139)

3. Lines 188-189: This sentence doesn’t seem to belong here, as it’s unrelated to MLST typing

Response: Thank you for the observation. The sentence has been deleted (Page 9, lines 198 - 199)

4. Lines 193-194: There are many more isolates from the Nyanza region, so it is unsurprising for some STs to be found there and nowhere else - please be sure to include the uneven distribution of isolates by region in your discussion.

Response: We agree that the with the observation. This has been clarified in the discussion section (Page 18, lines 438-442)

5. Lines 198-200: The wording here is somewhat unclear. An example of how this may be reworded is: Thirty-one (88.6%) isolates belonged to 26 novel NG-MAST STs, with 2 isolates each of STs and one isolate each of STs. Four isolates belonged to 3 known STs: ST-355 (n=?), ST-10134 (n=?), and ST-11752 (n=?). The most common ST was ST-19168, found in four isolates (11.4%) from Nyanza, all of which were also MLST ST-8133.

Response: This comment has been addressed under the major comments. Please see response to the second major comment above.

6. Lines 217-218: This sentence is unclear. Do you mean, “Two isolates, KNY_NGAMR20 and KNY_NGAMR23, from Nyanza shared the same NG-STAR, NG-MAST and MLST types”?

Response: The sentence has been rephrased (Page 11, lines 225-226)

7. Lines 222-223: Please add a denominator to your n values for each year. It is difficult to understand the context of the number of novel STs without knowing the number of samples in total per year.

Response: The denominators have been added as recommended and the whole section revised. (Page 11, line 231)

8. Line 330: Please change to “…identified a variety of…”

Response: Introduction paragraph of the discussion has been rephrased (Page 15, lines 351-357)

9. Line 337: Please change to “…Western Australia”

Response: Change effected (Page 15, line 357)

10. Lines 339-340: Previous studies have demonstrated that NG-MAST has a higher discriminatory power compared with MLST. Eg: https://pmc.ncbi.nlm.nih.gov/articles/PMC3811299/

Response: We agree with the reviewer’s observation and have revised the manuscript to acknowledge prior evidence supporting NG‑MAST’s superior discriminatory power (Page15, line 355-357)

11. Line 373: Please change to “…belong to a heterogenous population”

Response: Change effected (Page 18, line 435)

12. Lines 374-376: This sentence is unclear

Response: This section has been rewritten to enhance clarity (Page 18, line 430 - 433)

13. Line 466: AMR should be defined at the beginning of the article, not at the end

Response: The section has been rewritten and abbreviation deleted (Page 20, line 469)

Reviewer #2:

This manuscript describes the presence of different types of Neisseria gonorrhoeae (NG) strains in 4 regions in Kenya using genomic typing, after obtaining full genome sequences. It is interesting to see what has been circulating in this African country, since in the present databases mostly Western/ European types are described. In this respect it is interesting to see Figure 4 where the Kenyan sequences are in a tree with other global sequences of NG.

Although this reviewer recognizes that it is a formidable job to obtain whole genome sequences of NG strains, it is a severe limitation that only 35 strains were included. Kenya is a very large country with >50 million inhabitants, so this sample can hardly be representative for what is circulating. Another limitation is that only samples from 2013 to 2018 (5 years) were included which is at least 7 years ago from the present. These limitations should be clearly mentioned, also in the abstract.

Response: We acknowledge the reviewer’s concern regarding the limited number of isolates and the timeframe of sampling. Indeed, only 35 strains were included, collected between 2013 and 2018, which is a small dataset relative to Kenya’s population of over 50 million. This sample cannot be considered fully representative of the circulating gonococcal population nationwide. Furthermore, the isolates were obtained at least seven years prior to the present, and therefore may not reflect current strain diversity or resistance trends. These limitations have been clearly stated in the discussion section of the revised manuscript, to ensure transparency about the scope and generalizability of our findings (Page 19, line 438 - 442)

Some other major and minor points must be taken in consideration to further improve this manuscript.

Major Comments:

1. The Results are based on WGS data from 35 samples. In many lines the types are given in percentages. For example, in line 187/188 is stated that in 2015, 44.4% (CI….) of isolates displayed a novel MLST type. Please give the number of isolates instead of the percentage, since in 2015 this number was only 9 (line 223). Please adjust where applicable (also in lines 224 to 228 for example) to mention the absolute numbers of isolates and not (only) the percentages.

Response: The results section on sequence typing (Line 190) and temporal distribution of novel sequence types (line 228) have been rewritten to capture the recommended changes.

2. Figure 1: please present absolute numbers for novel types and not percentages. Again, the numbers are very small and not representative for the years.

Response: Change effected inside the graph as suggested (Fig 1)

3. The S1 Table is mentioned later in the text (line 157) then the S2 Table. Please reverse the S1 and S2 Table numbers so they will appear in the right order.

Response: S1 table is mentioned first in page 3, line 100. Then S2 in page 5, line 103

4. The sentence on ciprofloxacin resistance (lines 189-190) does not fit in the middle of the paragraph on MLST. Please mention this for example in the NG-STAR paragraph which deals with antibiotic resistance gene polymorphisms.

Response: Thank you for the observation. The sentence has been deleted (Page 9, lines 198 - 199)

5. In S3 Table can be seen that most (23/35) samples were derived from the Nyanza region. So, the finding that the two ST8133 types belonged to the Nyanza region is of no surprise. The other two each ‘region-based’ types ST-11365 and ST-13780 that were also from Nyanza is what can be expected by chance if so, many strains were from that region. So, the suggestion that there is clustering of MLSTs based on the region is not justified. The same comment goes for the NG-MAST types in lines 205-207. Please rephrase this, and avoid ‘region based’.

Response: We agree that the with the observation. This has been clarified in the discussion section (Page 18, lines 438-442)

6. S6 Table where relations are tested as ‘Group_vs_other’ is incomprehensible. I suggest to leave out this paragraph (lines 235 -243) and Table.

Response: The section on “association between typing schemes and antimicrobial resistance” and S6 Table have been deleted. A sentence explaining the relationship between typing schemes and AMR has been added to the NGSTAR phylogeny section (Page 14, lines 305-306)

7. In Figure 2 the Pub MLST IDs of the strains are difficult to read. You may consider to provide a supplementary table with numbers 1 to 35 in which per strain number each Pub MLST ID type number (MLST, NG-STAR- NG-Mast) is given. In that way all data on these strains will be more easily seen in the figure. Lines 268-270: I suggest that there actually is no Cluster 4; these strains are genetically too far apart.

Response: To improve clarity, we have provided a supplementary table (S6 Table) listing all 35 isolates, numbered sequentially, with their corresponding PubMLST IDs and assigned sequence types (MLST, NG-MAST, and NG-STAR). The sentence describing cluster 4 has been rephrased based on the observation made by the reviewer (Page 12 & 13, lines 274-276)

8. Line 285: The NG-STAR STs are mentioned in Figure 3. However, in this Figure the concatenated sequences of NG-STAR are shown. So, the NG-STAR type itself is not needed and redundant information. Please adjust.

Response: Thank you for this observation. We appreciate the concern regarding potential redundancy in Figure 3. While the phylogeny is indeed inferred from concatenated NG-STAR loci, we intentionally included NG-STAR ST labels to illustrate how specific STs cluster in relation to antimicrobial AMR profiles. This dual presentation allows readers to interpret both the evolutionary relationships based on concatenated loci and the epidemiological relevance of ST clustering, particularly in relation to resistance determinants. We believe retaining the NG-STAR STs in the figure provides added value by linking phylogenetic structure to practical surveillance markers.

9. The Discussion is very lengthy (5 pages) and contains both redundancies and textbook information. It should be reduced to 3 pages or less. For example: lines 374 - 376: this is a repetition of the results.; lines 429 (starting with ‘A small’) to 437 (up to ‘novelty’) is too general: text book stuff that can be deleted; lines 440-447: repetition of earlier text: delete

---

## [Decision Letter · Decision Letter 1]

29 Apr 2026

Genomic Diversity of Neisseria gonorrhoeae Isolates in Kenya Revealed by MLST, NG-MAST, and NG-STAR Typing

PONE-D-25-55705R1

Dear Dr. Kivata,

We’re pleased to inform you that your manuscript has been judged scientifically suitable for publication and will be formally accepted for publication once it meets all outstanding technical requirements.

Kind regards,

Sylvia Maria Bruisten, Ph.D

Academic Editor

PLOS One

Additional Editor Comments (optional):

This manuscript has been very well revised thereby improving its quality. It is now aceptable for publication in PlosOne.

Specifically the inclusion in the discussion of the recent publication of Mehta et al, 2025, has broadened the message and also the improvement of Figure 4 has substantially improved this paper.

Reviewers' comments:

Reviewer's Responses to Questions

**Comments to the Author**

1. If the authors have adequately addressed your comments raised in a previous round of review and you feel that this manuscript is now acceptable for publication, you may indicate that here to bypass the “Comments to the Author” section, enter your conflict of interest statement in the “Confidential to Editor” section, and submit your "Accept" recommendation.

Reviewer #2: All comments have been addressed

2. Is the manuscript technically sound, and do the data support the conclusions?

Reviewer #2: Yes

3. Has the statistical analysis been performed appropriately and rigorously? 

Reviewer #2: Yes

4. Have the authors made all data underlying the findings in their manuscript fully available?

Reviewer #2: Yes

5. Is the manuscript presented in an intelligible fashion and written in standard English?

Reviewer #2: Yes

6. Review Comments to the Author

Reviewer #2: The authors did a very good job in revising this manuscript. The paper has improved substantially and will contribute to the knowledge on circulating types of Neisseria gonorrhoeae in Kenya.

7. PLOS authors have the option to publish the peer review history of their article (what does this mean?). If published, this will include your full peer review and any attached files.

Reviewer #2: No

---

## [Editor Report · Acceptance letter]

PONE-D-25-55705R1

PLOS One

Dear Dr. Kivata,

I'm pleased to inform you that your manuscript has been deemed suitable for publication in PLOS One. Congratulations! Your manuscript is now being handed over to our production team.

Kind regards,

on behalf of

Dr. Sylvia Maria Bruisten

Academic Editor

PLOS One